# The Storm-Track Suppression over the Western North Pacific From a Cyclone Life-Cycle Perspective

Sebastian Schemm[1], Heini Wernli[1], and Hanin Binder[1]

[1]Institute for Atmospheric and Climate Science, Universitätstrasse 16, 8092 Zürich, Switzerland

**Correspondence:** sebastian.schemm@env.ethz.ch

**Abstract.** Surface cyclones that feed the western part of the North Pacific storm track and experience a midwinter suppression originate from three regions: the East China Sea ($\sim30°$N), the Kuroshio extension ($\sim35°$N), and downstream of Kamchatka ($\sim53°$N). In midwinter, in terms of cyclone numbers, Kuroshio (45%) and Kamchatka (40%) cyclones dominate in the region where eddy kinetic energy is suppressed, while the relevance of East China Sea cyclones increases from winter (15%) to spring (20%). The equatorward movement of the baroclinicity and the associated upper-level jet toward midwinter influences cyclones from the three genesis regions in different ways. In January, Kamchatka cyclones are less numerous, less intense and their lifetime shortens; broadly consistent with the reduced baroclinicity in which they grow. The opposite is found for East China Sea cyclones, which in winter live longer, are more intense, and experience more frequently explosive deepening. The fraction of explosive East China Sea cyclones is particularly high in January when they benefit from the increased baroclinicity in their environment. Again, a different and more complex behavior is found for Kuroshio cyclones. In midwinter, their number increases, but their lifetime decreases; on average they reach higher intensity in terms of minimum sea level pressure, but the fraction of explosively deepening cyclones reduces and the latitude where maximum growth occurs shifts equatorward. Therefore, the life cycle of Kuroshio cyclones seems to be accelerated in midwinter with a stronger and earlier but also shorter deepening phase followed by an earlier decay. Once they reach the latitude where eddy kinetic energy is suppressed in midwinter, their baroclinic conversion efficiency is strongly reduced. Together, this detailed cyclone life-cycle analysis reveals that the North Pacific storm-track suppression in midwinter is related to fewer and weaker Kamchatka cyclones and to more equatorward intensifying and then more rapidly decaying Kuroshio cyclones. The less numerous cyclone branch from the East China Sea partially opposes the midwinter suppression. The cyclones passing through the suppressed region over the western North Pacific do not propagate far downstream and decay in the central North Pacific. The behaviour of cyclones in the eastern North Pacific requires further analysis.

## 1 Introduction

Nakamura (1992) identified the contrasting intraseasonal cycles of the North Atlantic and North Pacific storm tracks in winter. While different measures of storm-track activity over the Atlantic experience a single peak in midwinter, the Pacific storm-track activity has two peaks, one in late autumn and another one in early spring (Nakamura, 1992). Over the North Atlantic, the seasonal cycle of the storm-track activity is broadly consistent with the seasonal cycle of mean baroclinicity, but this is not the

case over the North Pacific where the midwinter suppression of the storm-track activity occurs at the time of maximum surface baroclinicity and jet strength. The midwinter suppression affects a large number of eddy measures, such as the eddy heat and momentum fluxes and the eddy kinetic energy, as well as the baroclinic and barotropic conversion rates, whereas measures of the mean background flow, such as the monthly mean jet strength and the Eady growth rate are not affected by the suppression (Schemm and Schneider, 2018). The midwinter suppression is most pronounced in the upper troposphere and almost absent in the lower troposphere. Also, the total number of surface cyclones does not experience a suppression (Schemm and Schneider, 2018). The atypical intraseasonal cycle of baroclinic waves over the North Pacific has triggered considerable research during the last two decades. In the following, we give an overview of the current understanding of the midwinter suppression.

Factors that have been suggested to contribute to the midwinter suppression can be categorized into contributions from barotropic, baroclinic and upstream-seeding processes. Among the barotropic contributions are the increase of the horizontal wind shear near the jet and the deformation acting on baroclinic wave packets in a stronger and more narrow jet stream during midwinter (James, 1987; Nakamura, 1993; Harnik and Chang, 2004; Deng and Mak, 2005). However, in idealized simulations it was shown that such a barotropic governor mechanism is not symmetric in time around the suppression and the increase in the horizontal shear lags the onset of the suppression (Novak et al., 2020). With regard to upstream seeding processes, a reduction in the amplitude and frequency of upper-level eddies propagating from upstream into the North Pacific has been suggested to play a crucial role in the formation of the midwinter suppression (Penny et al., 2010, 2011, 2013), but this is strongly debated because baroclinic growth is decorrelated with the strength of upstream seeding (Chang and Guo, 2011, 2012). Moreover, a reduced midwinter suppression also occurs in simulations with the upstream Asian mountains removed (Park et al., 2010). The increase in the velocity of eddy propagation along the baroclinic zone and a reduced lifetime has also been shown to be insufficient to explain the suppression (Chang, 2001; Nakamura and Sampe, 2002). The reduction in the lifetime also occurs over the North Atlantic, which does not exhibit a suppression in most winters (Schemm and Schneider, 2018).

There is mounting evidence that the shift to a subtropical jet regime in the western North Pacific is essential for the formation of the midwinter suppression. During the shift to a subtropical jet regime, the subtropical thermally-driven jet increases its strength, shift equatorward and extends into the zonal direction. Idealized studies have shown that the transition to a subtropical jet regime is able to reproduce a realistic midwinter suppression and thus ruled out the absolute necessity of zonal asymmetries, such as upstream mountains, for its formation (Yuval et al., 2018; Novak et al., 2020). The importance of a subtropical jet regime for the formation of a storm-track suppression is reinforced by the fact that a mild suppression is also observed over the North Atlantic in years with strong subtropical jets (Penny et al., 2013; Afargan and Kaspi, 2017). Chang (2001) and Nakamura and Sampe (2002) already hinted at the potential key role of the subtropical jet and its meridional displacement relative to the low-level zone of highest baroclinicity, but the exact mechanism that reduces baroclinic growth remained unclear. Such a mechanism was suggested by Schemm and Rivière (2019), who showed that in the subtropical jet regime the ability of eddies to extract eddy energy from the mean baroclinicity is reduced because of a reduction in the baroclinic conversion efficiency. Eddies from the northern seeding branch (Chang, 2005) propagate more equatorward and towards the subtropical jet in midwinter and during this equatorward propagation they acquire a stronger than usual poleward tilt with height, which reduces the eddy efficiency because of a weaker alignment between the mean baroclinicity and the eddy heat flux (Schemm and

Rivière, 2019). Schemm and Rivière (2019) quantified baroclinic conversion for all upper-level eddies and surface cyclones in the western North Pacific. However, as shown in section 4 of this study, the surface cyclone tracks in this sector of the Pacific emerge from three different regions: (i) downstream of Kamchatka, (ii) over the Kuroshio extension, and (iii) over the East China Sea. So far, it is unclear if the suppression affects the cyclones from these genesis regions in a similar way.

In this study, we investigate midwinter changes in surface cyclone life cycles over the western North Pacific according to their genesis region. Surface cyclones are an important subcategory of the wide distribution of flow features collectively termed "eddies". Upper-level cyclonic eddies, some of them shallow, correspond to troughs. Once a trough interacts with a surface eddy, they mutually amplify (Hoskins et al., 1985, Fig. 21) and propagate in tandem poleward (Gilet et al., 2009; Rivière et al., 2012; Oruba et al., 2013). The combined system develops into a mature low-pressure system corresponding to a deep cyclonic
eddy. With surface cyclone tracks we thus identify particularly strong cyclonic eddies that play an essential role for the overall storm track climatology. Over the North Pacific, we expect to find different life-cycle characteristics in midwinter compared to November and March, because in midwinter the cyclones typically form on the poleward flank of a strong subtropical jet, whereas in November and March they usually develop on the equatorward flank of a more poleward located jet. Schemm and Schneider (2018) have already shown that for the entire North Pacific the lifetime of surface cyclones decreases. Here we study
in detail all surface cyclones that affect the region of the midwinter suppression in the western North Pacific between October and April and quantify their frequency, lifetime, intensity, baroclinic conversion rates, and other characteristics according to their genesis region. This approach will serve to address the following questions:

- What is the relative contribution of different genesis regions to the surface cyclone frequency in the region affected by the midwinter suppression?

- Are there any differences in the character of the surface cyclones of different origin between midwinter and the shoulder months? For example, how does their number, lifetime and time to maximum intensity vary during the cold season?

- Is the suppression of the baroclinic conversion during midwinter equally strong for cyclones of different origin?

To answer these questions, we use an object-based surface cyclone tracking algorithm and evaluate baroclinic conversion rates obtained from bandpass-filtered data along individual cyclone tracks. With this approach, we combine two complementary
perspectives on storm track dynamics.

Our study is organized as follows. In section 2 we introduce the used data and methods. In section 3 we describe the midwinter evolution of eddy kinetic energy (EKE) over the North Pacific and define specific target regions characterized by an increase and decrease in EKE during winter, respectively. The surface cyclone tracks and the genesis regions of cyclones that propagate through the target regions are presented in section 4. Section 5 presents a detailed analysis of changes in different
life cycle characteristics. Baroclinic conversion along cyclone tracks of different origin are studied in section 6. We conclude our study in section 7.

## 2 Data and Methods

The analysis period is October to April 1979–2018. All diagnostics rely on 6-hourly ERA-Interim data that are interpolated to a $1°$ grid. ERA-Interim is publicly available for download via ECMWFs archive at https://apps.ecmwf.int/datasets/.

### 2.1 Surface-cyclone tracks and surface cyclogenesis

For the identification and tracking of surface cyclones, we make use of the algorithm introduced by Wernli and Schwierz (2006) and refined by Sprenger et al. (2017). The detection of surface cyclones is based on a contour search in the mean sea level pressure (SLP) field at intervals of 0.5 hPa. To obtain a cyclone mask at each time step, all grid points inside the outermost closed contour, which must not exceed 7500 km in length, are labelled with 1, all others with 0. The obtained binary cyclone fields are used to compute cyclone frequencies. Cyclone centers are defined as the grid point with minimum SLP inside the outermost closed contour. The cyclone centers are tracked using 6-hourly cyclone center positions, and a track is accepted if it exists for a period of at least one day. The first time step along each track is defined as the genesis time step and the SLP minimum defines the genesis location. The algorithm contributed to the cyclone identification and tracking intercomparison project of Neu et al. (2013).

### 2.2 Baroclinicity and baroclinic conversion

The background baroclinicity and the corresponding baroclinic conversion are defined based on tendency equations for eddy kinetic and available potential energy (Lorenz, 1955; Orlanski and Katzfey, 1991; Chang, 2001). A detailed derivation of both tendency equations using 10-day high-pass-filtered input data is given in Schemm and Rivière (2019). The baroclinic conversion to eddy energy, the sum of eddy kinetic and eddy available potential energy, is the scalar product between the eddy heat flux $\frac{1}{\sqrt{S}}\theta'\mathbf{v}'$ and the background baroclinicity $-\frac{\nabla\overline{\theta}}{\sqrt{S}}$,

$$B_{conv} = -\frac{1}{S}\theta'\mathbf{v}' \cdot \nabla\overline{\theta} \tag{1}$$

where $\mathbf{v}'$ denotes the high-pass-filtered horizontal wind, $\overline{\theta}$ the low-pass-filtered potential temperature, and $S$ the static stability in pressure coordinates $S = -h^{-1}\frac{\partial\theta_R}{\partial p}$. The reference potential temperature $\theta_R$ is computed from monthly mean data and $h$ denotes the scale height. Background baroclinicity is defined as the horizontal gradient of the low-pass-filtered potential temperature divided by the static stability, $\boldsymbol{B} = -\frac{\nabla\overline{\theta}}{\sqrt{S}}$. The background baroclinicity is closely related to the Eady growth rate (Lindzen and Farrell, 1980). The scalar product that defines the baroclinic conversion can further be decomposed into contributions from the background baroclinicity and the baroclinic conversion efficiency, which is a measure for how efficiently cyclones convert the baroclinicity into eddy energy. The baroclinic conversion efficiency is connected to the vertical tilt of the growing cyclone and is maximized if the vertical tilt is such that the eddy heat flux aligns with the mean baroclinicity – for more details see Schemm and Rivière (2019). We base our analysis on the conversion rates at the 500 hPa level. This is a pragmatic choice, because the suppression increases in amplitude with altitude, while baroclinic conversion is largest in the

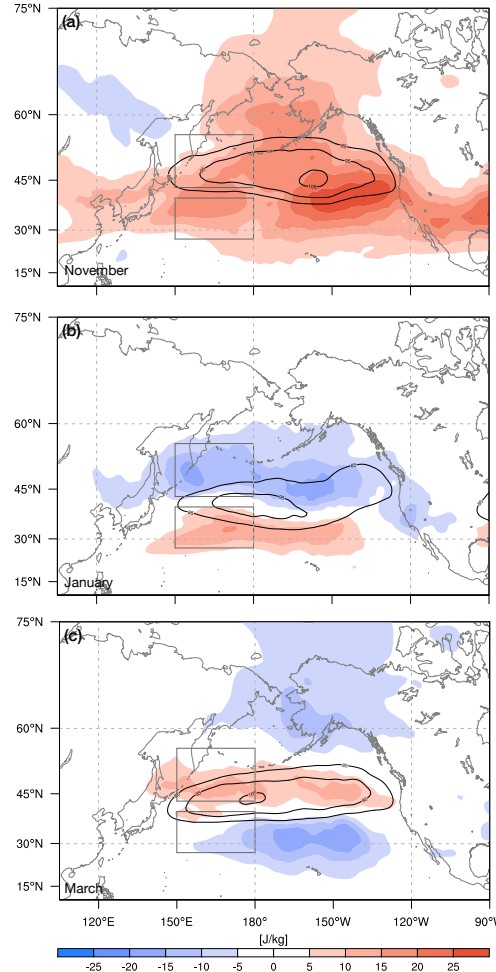

**Figure 1.** Mean EKE at the 500 hPa level (black contours at 85, 95, and 105 J kg$^{-1}$) and corresponding change relative to the corresponding previous month (color shading) for (a) November, (b) January and (c) March. Additionally shown are two target regions (gray boxes) that are used for the detailed diagnostics of surface cyclone tracks throughout this study.

lower troposphere (Schemm and Schneider, 2018). At the 500 hPa level, the suppression of baroclinic conversion is a well-marked feature.

## 3  EKE of the North Pacific storm track in midwinter

This section recapitulates the seasonal cycle and transition of the North Pacific storm track as seen in EKE at 500 hPa. In November, EKE increases, relative to October, across the entire North Pacific (red shading in Fig. 1a), with the maximum increase over the eastern North Pacific near 150°W. In January, EKE reduces compared to December in an elongated band

**Table 1.** (First column) Total number of surface cyclones in the northern target region and number of cyclones per day in parenthesis (1980–2018). (Second column) Fraction and number (in parenthesis) of Kamchatka, (third column) Kuroshio and (fourth column) East China Sea cyclones. The northern target region is shown in Fig. 1. Kuroshio and East China Sea cyclones have their genesis south of 45°N and east of 135°E and west of 135°E, respectively.

| | Cyclones in target region | Kamchatka | Kuroshio | East China Sea |
|---|---|---|---|---|
| November | 507 [0.43] | 45 % [229] | 41 % [207] | 14 % [71] |
| January | 516 [0.43] | 40 % [206] | 46 % [236] | 14 % [74] |
| March | 527 [0.44] | 39 % [208] | 39 % [204] | 22 % [115] |

across the Pacific north of ∼43° (blue shading in Fig. 1b). Equatorward of this latitude, however, EKE increases (red shading in Fig. 1b). This dipole pattern in EKE tendency results from the equatorward shift of the North Pacific jet during midwinter.
Absolute values of EKE (black contours in Fig. 1b) are reduced compared to those in November, which is due to the stronger EKE reduction poleward of ∼43°N compared to the simultaneous increase equatorward [see Fig. 1 in Schemm and Schneider (2018) for a more detailed discussion of month-to-month EKE variations]. From February to March, EKE increases again in a meridionally confined band between ∼40° and 50°N and decreases equatorward of ∼40°. Based on theses patterns of intraseasonal changes in EKE, we select in the following surface cyclones that propagate through one of the regions with a
midwinter (December to January) EKE decline or increase, respectively. The two target regions are indicated as gray boxes in Fig. 1 and are located at the entrance of the storm track (EKE maximum). One of our aims is to assess if and how the EKE tendency dipole seen in Fig. 1b is linked to characteristics of the surface cyclone tracks. We focus in particular on cyclones that propagate through the northern target region, which exhibits a decline in EKE during midwinter and is therefore essential for understanding the midwinter suppression phenomenon.

**4  Surface-cyclone view on the North Pacific storm track in midwinter**

**4.1  Surface cyclogenesis**

Consideration is first given to cyclogenesis associated with surface cyclone tracks that propagate through the northern target region, i.e., the region where EKE decreases during midwinter (Fig. 1b). To this end, we extract all tracks for which the cyclone center is insight the target region for at least one time step. In November, these surface cyclones originate from two preferred
regions (Fig. 2a). The first region is located over the Kuroshio extension (near 35°N) and the second region downstream of the Kamchatka Peninsula (near 53°N). This pattern and the number of events are fairly similar in January (Fig. 2c). In March

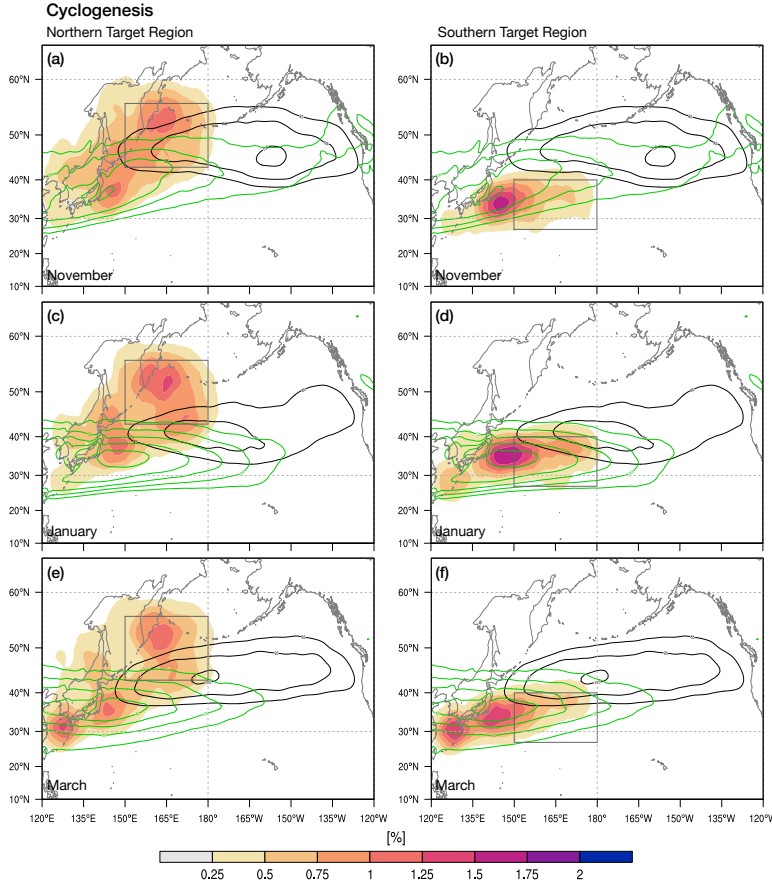

**Figure 2.** Cyclogenesis frequency (color shading; %) for surface cyclone tracks that propagate through the northern (left column) and southern (right column) target regions (shown as a gray box) for (a,b) November, (c,d) January and (e,f) March. Additionally shown are EKE (black contours; 85, 95, and 105 $\mathrm{J\,kg^{-1}}$) and baroclinicity (green contours; 25 to 45 by steps of $5\times10^{-6}\mathrm{s^{-1}}$) at the 500-hPa level

(Fig. 2e), a third cyclogenesis region emerges southwest of Japan at over the East China Sea (near 30°N), while the other two cyclogenesis regions retain fairly similar frequencies. For cyclones propagating through the target region of the midwinter suppression, there is no signal of a suppression in the genesis and therefore number of these cyclones (Tab. 1).

The southern target region, i.e., the region where EKE increases during midwinter (Fig. 1b), is fed exclusively by surface cyclones with genesis over the Kuroshio extension during November (Fig. 2b). In January, a second but weaker cyclogenesis region emerges southwest of Japan over the East China Sea. In March (Fig. 2f), the two genesis regions exhibit similar cyclogenesis frequencies and contribute equally to the cyclone tracks in the southern target region. Notably, also for this region there is no midwinter suppression in the cyclone frequency. This result is in agreement with the findings of Schemm and Schneider

(2018) that the suppression is connected to a reduction in the cyclones' intensity rather than frequency.

Surface cyclones over the North Pacific are known to be triggered by two upper-level seeding branches: a northern branch over Siberia and a southern branch along the subtropical jet across southern Asia (Chang, 2005). We briefly report about the upper-level seeding associated with the three preferred regions of surface cyclogenesis (Fig. 2) by means of lagged 300 hPa geopotential anomalies. We find that Kamchatka and Kuroshio cyclogenesis is triggered by waves entering the North Pacific from the northern seeding branch over Siberia (Supplementary Figure S1), which, in January, have a more equatorward propagation direction, in agreement with Schemm and Rivière (2019). After surface cyclogenesis, the upper-level wave packet retains its overall more zonal orientation (Supplementary Figure S1). Cyclogenesis over the East China Sea is associated with the southern seeding branch and an upper-level trough downstream of the genesis location (not shown). This behavior was already recognized by Chang (2005), who noted that "cyclogenesis for these cases is probably not triggered by the [upper-level] wave packet" (Chang, 2005, p.1998). The genesis of these cyclones seems to be connected to a bottom-up development, as is the case for diabatic Rossby waves (e.g., Boettcher and Wernli, 2013).

## 4.2  Relative surface cyclone frequencies

In the previous section, we showed that surface cyclogenesis downstream of Kamchatka, over the Kuroshio extension and, in late winter, over the East China Sea, contribute to the surface cyclone tracks in the northern target region, where EKE exhibits a midwinter suppression. To study the relative importance of the tracks generated in the different genesis regions for the total cyclone frequency in the target region and elsewhere, we group the cyclone tracks into two categories. The first contains tracks with genesis over the Kuroshio or the East China Sea, which enter the target region from the south. The second category contains tracks with genesis near Kamchatka that propagate through the target region. Next, we compute cyclone frequency fields for the two categories at every grid point and divide them by the total cyclone frequency field obtained from all cyclone tracks. The obtained relative contributions are shown for November, January and March in Fig. 3. During all months, Kamchatka cyclones contribute up to 40–50% along the poleward side of the target region and only 10–20% along the equatorward side. The relative contribution of cyclone tracks entering the target region from the south is 80–90% along the equatorward side of the box and decreases towards higher latitudes to 50–60% along the poleward side of the target region. The relative contributions indicate the tendency of the selected cyclone tracks to propagate poleward. For example, less than 20% of the cyclone tracks propagated from the Kuroshio region across the Pacific and into the Gulf of Alaska. Figure 3 further corroborates that the midwinter suppression is related to a change in the characteristics of these cyclones and not in their frequencies. Also, we cannot focus on either Kuroshio or Kamchatka cyclones only, because they both contribute substantially to the total cyclone frequency in the target region. In the following, we study statistics of cyclone characteristics for the different genesis regions in greater detail.

## 5  Detailed characteristics of surface cyclone life cycle

In the previous section, we showed that the northwestern Pacific surface storm track is fed by three preferred cyclogenesis regions: (i) East China Sea, (ii) Kuroshio and (iii) Kamchatka; and neither of the three exhibits a midwinter suppression in

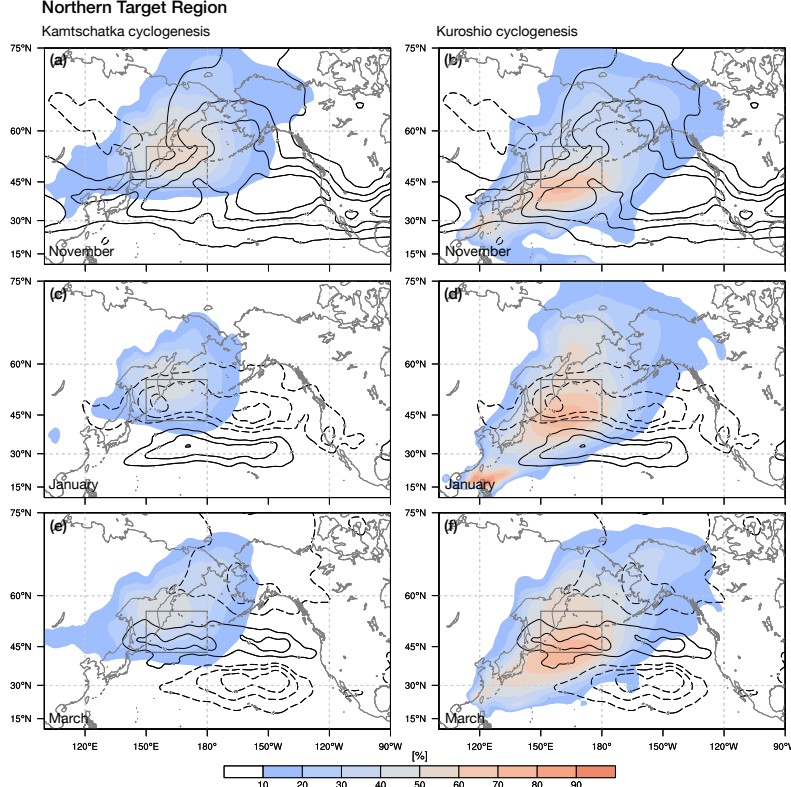

**Figure 3.** Relative contributions (color shading; %) of (a, c and e) Kamchatka cyclones and (b, d and f) Kuroshio combined with East China Sea cyclones to the total surface cyclone frequency in the northern target region (gray box). The contours show the change in EKE relative to the corresponding previous month (solid lines are for positive values, and dashed lines for negative values; -20 to 20 $J\,kg^{-1}$ by steps of 5 $J\,kg^{-1}$).

terms of cyclogenesis frequency (Fig. 2). Next, we investigate several life cycle characteristics. As in the previous sections, our focus is on the northern target region (gray box in Fig. 1), where EKE decreases during midwinter.

The lifetime from genesis to lysis of Kamchatka and Kuroshio cyclones is shortest in January and larger in November and March (Tab. 2). This is what one might expect from the strong midwinter jet and the fact that also over the North Atlantic lowest cyclone lifetimes are observed during midwinter (Schemm and Schneider, 2018). However, it could also result from the fact that these poleward propagating cyclones leave the more equatorward located baroclinic zone earlier. Finally, the lifetime of East China Sea cyclones increases from November to March.

The time to maximum deepening since cyclogenesis is shortest during midwinter, independent of the cyclogenesis region (Tab. 3). Hereby, maximum deepening is measured in Bergeron, which is the 24-hour change in sea level pressure along a cyclone track normalized to 60°N (Sanders and Gyakum, 1980). Because the cyclogenesis regions exhibit almost no variations in terms of their exact location (Fig. 2), the reduced time to maximum deepening for Kuroshio and East China Sea cyclones

**Table 2.** Mean lifetime (hours) of surface cyclones passing through the northern target region according to their genesis regions.

|           | Kamchatka | Kuroshio | East China Sea |
|-----------|-----------|----------|----------------|
| November  | 77        | 98       | 117            |
| January   | 62        | 86       | 132            |
| March     | 70        | 91       | 172            |

**Table 3.** Mean time to maximum deepening since genesis (hours) of cyclones passing through the northern target region according to their genesis regions.

|           | Kamchatka | Kuroshio | East China Sea |
|-----------|-----------|----------|----------------|
| November  | 35        | 29       | 43             |
| January   | 26        | 21       | 39             |
| March     | 31        | 23       | 42             |

could result from the more equatorward location of the zone of highest baroclinicity. These poleward propagating systems eventually leave the baroclinic zone earlier in midwinter, which explains the reduced lifetime and the shorter time to maximum deepening. For Kuroshio and East China Sea cyclones, the mean latitude where maximum deepening occurs is therefore also shifted by around 2° equatorward in January compared to November (not shown), which is in agreement with the earlier deepening.

With regard to the minimum sea level pressure as a measure of the storm intensity, Kamchatka cyclones become less intense from November to March (Tab. 4). East China Sea cyclones are most intense during midwinter, but they contribute only by 22 % to the total cyclone number (Tab. 1). Kuroshio cyclones are also most intense in January, but the change in minimum SLP between November and January is small. The equatorward movement of the baroclinic zone in midwinter seems to be beneficial for the intensification of East China Sea cyclones in January. Kamchatka cyclones, however, become less intense, a result that can, at least qualitatively, be expected from the equatorward retreat of the baroclinic zone. For Kuroshio cyclones the situation is complex. While there is a weak reduction in the minimum SLP from November to January, the fraction of life cycles that satisfy the Sanders and Gyakum (1980) criterion for explosive deepening, known as "bomb cyclogenesis"[1], is

---

[1]A change in SLP larger than 24 hPa within 24 hours normalized to 60°N (Sanders and Gyakum, 1980).

**Table 4.** Mean minimum sea level pressure of surface cyclones passing through the northern target region according to genesis regions and, in parenthesis, the fraction of cyclones satisfying the criterion for "bomb cyclogenesis" (deepening larger than 24 hPa within 24 hours normalized to 60°N.)

|          | Kamchatka      | Kuroshio       | East China Sea |
|----------|----------------|----------------|----------------|
| November | 979.4 [21 %]   | 975.8 [49 %]   | 975.7 [63 %]   |
| January  | 982.9 [7 %]    | 973.6 [42 %]   | 967.2 [76 %]   |
| March    | 987.3 [7 %]    | 979.0 [43 %]   | 971.4 [65 %]   |

reduced in midwinter. In contrast, for the East China Sea cyclones the bomb fraction peaks in January, which is in agreement with the increase in the background baroclinicity. Kuroshio cyclones thus appear to deepen rapidly in a short time period (Tab. 3), in agreement with the midwinter peak in baroclinicity, but after reaching their strongest intensity they also decay rapidly, as indicated by the shortest life time during midwinter (Tab. 2). This suggests that for Kuroshio cyclones the peak in baroclinic conversion occurs earlier during their life cycle and the short deepening is also more intense, but thereafter they move relatively soon out of the zone of high baroclinicity, resulting in less intense cyclones at higher latitudes. This could also explain the dipole pattern in EKE shown in Fig. 1b, because in January approximately 50% of all cyclones tracks that propagate through the northern target region also propagate through the southern target region. Thus, to better understand the intensification, in the next section we investigate baroclinic conversion first over the two target regions from an Eulerian viewpoint and afterward along the different tracks using feature-based cyclone tracking.

## 6  Baroclinic conversion and its relationship with surface cyclone tracks

### 6.1  Baroclinic conversion over target regions (Eulerian perspective)

EKE has a baroclinic and barotropic source, and both are known to be affected by midwinter suppression (Schemm and Schneider, 2018). In general, however, the dominant source of EKE is baroclinic conversion. In the following, we first diagnose variations in baroclinic conversion and its link to surface cyclones in the northern and southern target regions on synoptic time scales. The daily mean values of baroclinic conversion, averaged over the northern and southern target regions (see Fig. 1), are shown in Fig. 4 for November, January and March. Baroclinic conversion often peaks at regular intervals of 6-10 days (see the zoom-in for November 2009 and January and March 2010 in the right panels of Fig. 4). These baroclinic conversion bursts result mostly, but not exclusively, from the propagation of deep synoptic systems through the target region. In the northern target region in November 2009 (black contours in the top right panel of Fig. 4), the first and third bursts (labelled "1" and "3"

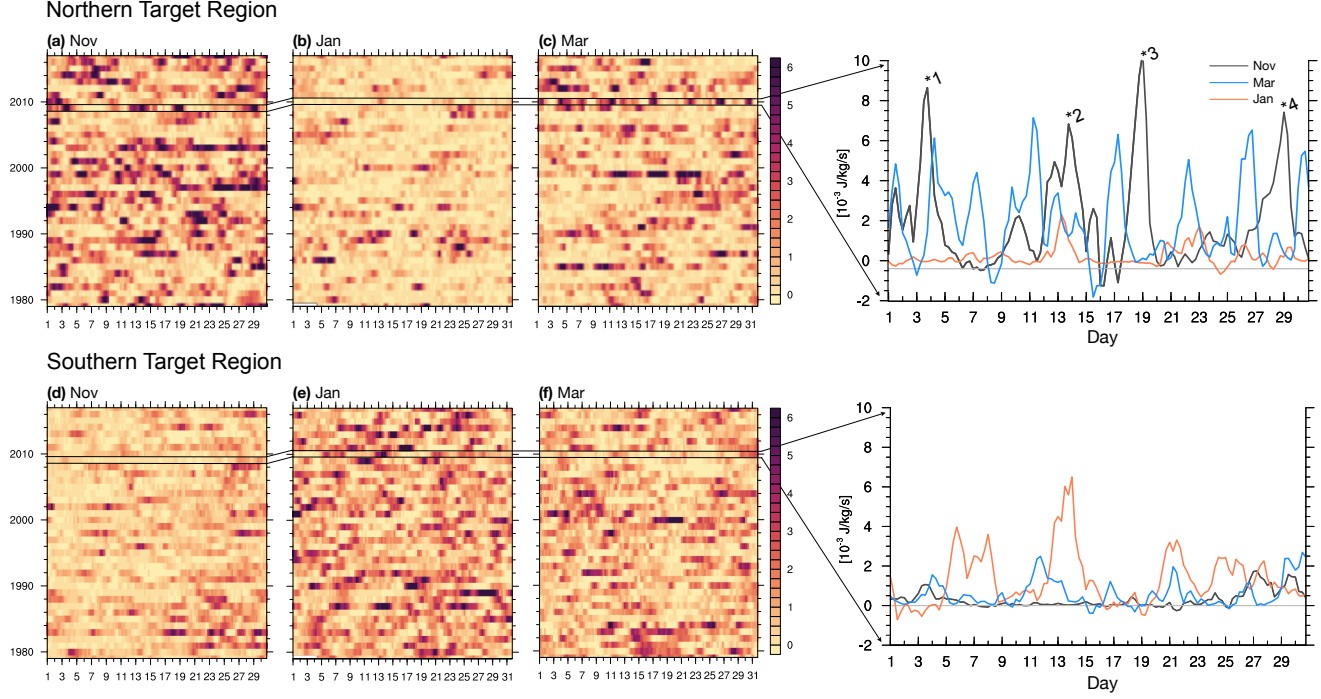

**Figure 4.** Baroclinic conversion at 500 hPa ($10^{-3}$ J kg$^{-1}$ s$^{-1}$) averaged over the (a, b, and c) northern and (d, e, and f) southern target regions shown in Fig. 1 for (a and d) November, (b and e) January and (c and f) March for the period 1979–2018. Attached on the right side is a zoomed-in image of the individual time series of daily mean values for November, January, and March 2009/2010.

in Fig. 4) are associated with Kuroshio cyclones, while the fourth one is associated with a Kamchatka cyclone that propagates north of the target region but still affects a broader region around it. The second burst (13–15 November) is not associated with a surface cyclone, but with a jet streak development at the edge of an upper-level trough. In November and March (black and blue contours in the top right panel of Fig. 4), the amplitude of the bursts exceeds those in January by a factor of 2–3 (orange contour). The opposite is found in the southern target region, where the baroclinic conversion bursts in January exceed those in November and March. However, the monthly differences in the southern target region are smaller than in the northern target region. Furthermore, the difference between January and March is less clear in the southern target region.

The above findings suggest that the characteristics or the synoptic systems that propagate through the two target regions shown in Fig. 1 clearly differ between the three months. In January, the associated baroclinic conversion is lower than in November and March in the northern target region, and vice versa for the southern region. There are several possible explanations for this behavior, for instance: (i) in midwinter, baroclinic conversion is reduced along the entire life cycle of cyclones in the northern region, or (ii) the life cycles of those cyclones that propagate through both target regions, which are about $\sim 50\%$ of all cyclones that enter the northern target region from the south, have an earlier baroclinic conversion peak (in the southern target region) and reduced baroclinic conversion later in the northern target region. As we show below, the first scenario ap-

plies to Kamchatka cyclones and the second one to Kuroshio cyclones. But first, we explore how baroclinic conversion changes during days when a cyclone propagates through the northern target region.

## 6.2 Baroclinic conversion in the northern target region associated with surface cyclones

In order to quantify the contribution of surface cyclones to the climatological monthly mean baroclinic conversion in the northern region, we split all days into cyclone and non-cyclone days using the surface cyclone tracks. Thereby, we essentially separate separate deep eddies that extend throughout the troposphere, like mature cyclones, and shallow diabatically maintained low-level cyclonic eddies, like diabatic Rossby waves, from upper-level shallow eddies, like troughs or ridges. Anticyclonic eddies are also excluded. Technically, a surface cyclone track may propagate outside of the northern target region but nevertheless affect the baroclinic conversion inside the target region. We therefore define a cyclone day as a time step when 25 % of the northern target region is covered by a cyclone mask (see section 2 for details). This results in about 50 % cyclone and 50 % non-cyclone days.

Baroclinic conversion is, as expected, larger during cyclone days compared to non-cyclone days (Fig. 5), but baroclinic conversion is not zero during non-cyclone days. Only in January, the median of the baroclinic conversion distribution is near zero for non-cyclone days. The fact that baroclinic conversion is not zero during non-cyclone days can be explained by baroclinic conversion related to an upper-level trough propagating over the northern target region that is not accompanied by a surface cyclone, as is the case for the second burst in Fig. 4 (upper right panel). As clearly shown in Fig. 5, the midwinter suppression affects baroclinic conversion during cyclone and non-cyclone days. Yet, the two distributions differ significantly from each other, in particular in January during midwinter suppression. To test this statistically, we compute 10'000 distributions, each of which consists of randomly selected cyclone and non-cyclone days with replacement. Each randomized distribution is of equal size as the original cyclone-day distribution. For each randomized distribution, we compute the mean baroclinic conversion, and from the 10'000 mean values the 97.5th and 2.5th percentiles, which are shown as red confidence intervals in Fig. 5. The mean baroclinic conversion values of the cyclone (non-cyclone) day distribution is above (below) the 97.5th (2.5th) confidence intervals in each month. We therefore conclude that the two distributions significantly differ from each other and from a randomized selection. Based on Fig. 5, we conclude that baroclinic conversion in the target region is reduced in midwinter both during cyclone days and non-cyclone days. However, since the baroclinic conversion during cyclone days is higher than during non-cyclone days, the cyclone days contribute in absolute terms more to the total baroclinic conversion in the Pacific storm track. Nevertheless, the relative contribution to the suppression is fairly similar. The average baroclinic conversion on a cyclone day reduces from 17 to $6 \times 10^{-4}$ J kg s$^{-1}$ and on a non-cyclone day from 13 to 3 $\times 10^{-4}$ J kg s$^{-1}$ and the number of days in each category is close to 50 % (Fig. 5). The separation into cyclone days and non-cyclone days is not clear cut. We defined non-cyclone days as those days during which the target region is covered by less than 25 % with a cyclone mask. Baroclinic conversion on non-cyclone days thus might still be associated with surface cyclones in close proximity of the target region. In a next step, we investigate in more details those cyclone tracks that enter the northern target region from the south and compare the baroclinic conversion along these tracks before and after entrance.

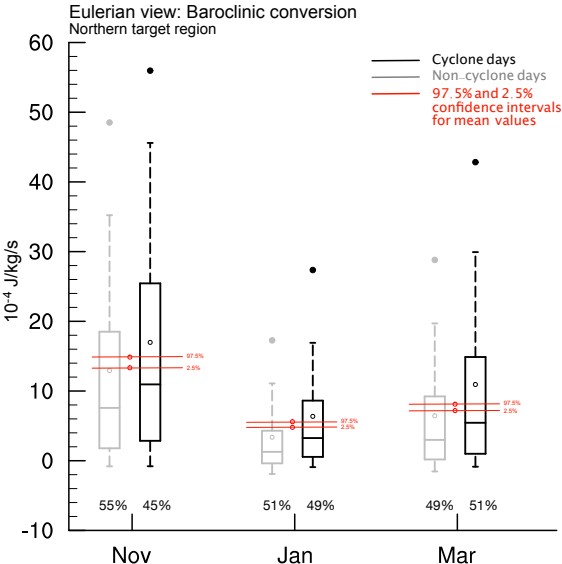

**Figure 5.** Box-and-whisker diagram of baroclinic conversion at 500 hPa ($10^{-4}$ J kg$^{-1}$ s$^{-1}$) averaged over the northern target region (gray box in Fig. 1) for days (black) with and (gray) without a surface cyclone affecting the target region (referred to as cyclone and non-cyclone days). Whiskers span between the 10th and 90th percentiles and the box spans the 25th to 75th percentile range. Filled dots indicate the 95th percentile. Open circles indicate the mean value and horizontal lines the median value. The 2.5 and 97.5 confidence intervals of a statistical test (see text for details) are shown in red. Percentage values at the bottom indicate the fraction of days in each sample.

## 6.3 Baroclinic conversion along cyclone tracks outside and inside the northern target region (Lagrangian perspective)

Cyclones that feed the northern target region are Kamchatka cyclones, with genesis inside the target region, and Kuroshio and East China Sea cyclones, which enter the target region from the south. We therefore group all time steps along Kuroshio and East China Sea cyclone tracks into two periods, before and after entering the northern target region. The idea is to see whether the maximum in baroclinic conversion occurs earlier during the life cycle in January, as suggested in section 5 and based on Tab. 3, and therefore outside the northern target region. In the following, we discuss box-and-whisker diagrams of baroclinic conversion, the background baroclinicity (Fig. 6a) and of the baroclinic conversion efficiency (Fig. 6 b) separately for cyclones that enter the target region from the south (Kuroshio and East China Sea cyclones) for time steps before and after entering the target region, and for Kamchatka cyclones, which reside inside the target region. For all cyclones, baroclinic conversion and its efficiency are averaged within a 1000 km radius around the cyclone center.

Before entering the northern target region from the south, baroclinic conversion along cyclone tracks is larger in January than in November and March (black boxes in Fig. 6). The distribution of baroclinic conversion outside of the target region exhibits a seasonal cycle that is qualitatively in agreement with the seasonal cycle of the mean baroclinicity equatorward of

the target region. The difference between January and March is small, which is a result of the increasing influence of East China Sea cyclone tracks towards late winter and early spring. East China Sea cyclones deepen on average more rapidly than
295 Kuroshio cyclones (Tab. 4) and they are particularly frequent in March (Fig. 2 and Table 1).

After entering the northern target region, baroclinic conversion associated with Kuroshio and East China Sea cyclones is reduced in January compared to November and March, which reflects the midwinter suppression (gray boxes in Fig. 6). Kuroshio cyclones spend most of their life cycle in the northern target region (percentage of time steps in each category is shown below each box in Fig. 6). In January, the fraction of time steps outside the northern target region is lower than in November
and March suggesting that in January cyclones propagate faster poleward and hence out of the zone of high baroclinicity as originally hypothesized by Nakamura (1992). On their way poleward, they also become less efficient in converting the mean baroclinicity (Fig. 6b). The reduction of baroclinic conversion in the northern target region in January occurs despite the fact that the mean baroclinicity along the tracks of Kuroshio cyclones is only marginally reduced compared to other months (horizontal red bars on top of the gray box-and-whiskers in Fig. 6). Equatorward of the target region, the baroclinic conversion
efficiency in January is higher compared to November, but once Kuroshio cyclones have entered the target region the conversion efficiency reduces and is lower in January. In November, the mean efficiency does even increase when Kuroshio cyclones enter the northern target region. The conversion budget discussed in Schemm and Rivière (2019) showed that the reduced baroclinic conversion in this region results indeed from both, a reduction in the mean baroclinicity and in the conversion efficiency, with the conversion efficiency making the larger contribution to the reduction. Overall, the here presented results indicate that
Kuroshio cyclones in January deepen rapidly equatorward of the target region and their growth is even stronger compared with November and March, but on their way poleward their conversion efficiency decreases, in agreement with the dipole anomaly seen in Fig. 4 in Schemm and Rivière (2019). The stronger growths in January equatorward of the target region appears to accelerate the life cycle and the cyclones seem to reach earlier during the life cycle a stage when they become less efficient in converting the mean baroclinicity into eddy energy.

For Kamchatka cyclones, baroclinic conversion is reduced in midwinter compared to the shoulder months (blue boxes in Fig. 6), in agreement with the reduced baroclinicity and reduced efficiency (Fig. 6b). The reintensification of Kamchatka cyclones during March occurs despite no notable change in the mean baroclinicity (horizontal red bars on top of the blue box-and-whiskers in Fig. 6), a finding that points again towards an increase in the baroclinic conversion efficiency as an important moderating process. Kamchatka cyclones contribute with about 40 % to the storm track over the northern target region and
their weakening is therefore an important contribution to the suppression.

In summary, maximum baroclinic conversion along the surface cyclone tracks with genesis over the Kuroshio extension is climatologically largest in January, but occurs equatorward of the northern target region and therefore earlier during the cyclone life cycle. The larger conversion is thus in agreement with an overall higher mean baroclinicity over the North Pacific, but the equatorward shift of the strengthened baroclinic zone is causing an earlier intensification, because the zone shifts towards
the preferred region of Kuroshio cyclogenesis. Kuroshio cyclones not only leave the zone of highest baroclinicity faster in midwinter, but on their way poleward they also become less efficient in converting the mean baroclinicity into eddy energy.

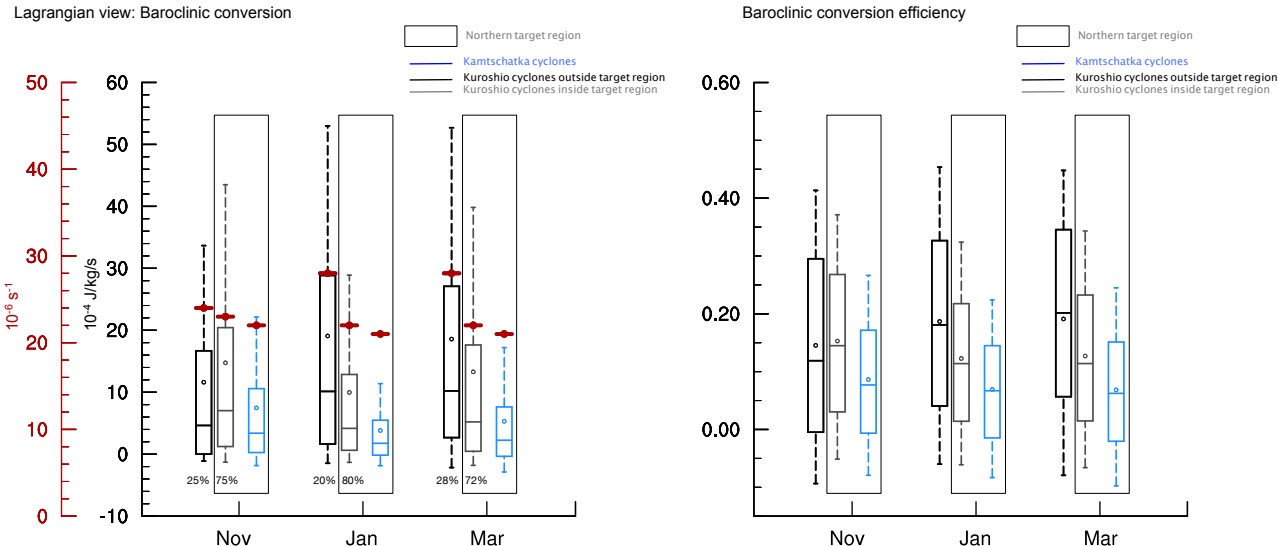

**Figure 6.** Box-and-whisker diagram for (a) baroclinic conversion at $500\,\text{hPa}$ ($10^{-4}\,\text{J}\,\text{kg}^{-1}\,\text{s}^{-1}$) averaged within a radius of $1000\,\text{km}$ around the surface cyclone centers of Kuroshio and East China Sea cyclones, before (black) and after (gray) entering the northern target region, and for Kamtschatka cyclones (blue). Additionally shown are the mean background baroclinicity along the tracks (red horizontal lines) and the percentage of time steps before and after entering the target region. (b) Similar as on the left but for the baroclinic conversion efficiency.

For Kamtschatka cyclones, the change throughout the cold season is fairly well in agreement with what must be expected from a reduction in the background baroclinicity during midwinter and an increase of it in early spring.

## 6.4 Baroclinic conversion along cyclone tracks in a target region that shifts with the maximum in monthly mean baroclinic conversion

In the previous sections, attention was given to baroclinic conversion in a target region that is centered on the location of the maximum reduction in EKE during midwinter. In this section, we explore baroclinic conversion associated with cyclone tracks that propagate through a target region that shifts with the location of the maximum in monthly mean baroclinic conversion (black target box in Fig. 7). The maximum is thoughout the winter located over the western North Pacific and shifts equatorward in January. The target box, which has the same latitudinal and longitudinal range in each month, is shifted latitudinally only. Baroclinic conversion reduces north of approximately $41°N$ in January and increases south of it (blue shading in Fig. 7a). Next, in line with our previous analysis, we identify all surface cyclone tracks that propagate into this target region from upstream. Fig. 7 shows the mean position of the selected tracks (red lines in Fig. 7). The black dots along this mean track indicate the mean location of cyclogenesis, the location of maximum deepening (defined as the largest 6-hourly reduction in mean SLP), the location of maximum intensity (defined as the minimum in SLP) and the mean location of cyclolysis. The first result is that

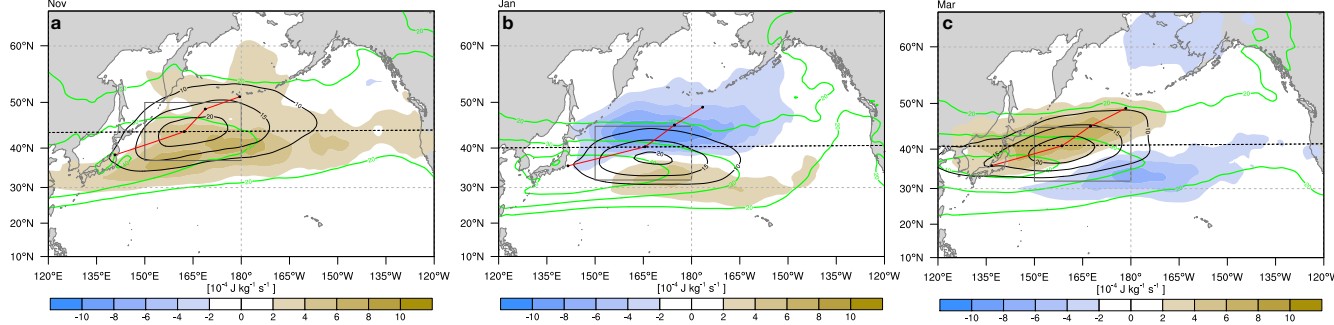

**Figure 7.** Monthly mean baroclinic conversion at 500 hPa (black contours, $10^{-4}$ J kg$^{-1}$ s$^{-1}$), its change relative to the previous month (color shading) and a target region (black box) that shifts with the maximum in the mean baroclinic conversion for (a) November (150°E–180, 37–50°N), (b) January (150°E–180, 32–45°N) and (c) March (150°E–180, 33–46°N). The red line indicates the mean location of cyclone tracks that propagate into the target region from upstream. The black dots along the mean track indicate the mean location of cyclogenesis, the location of maximum deepening (6-hourly SLP change), the location of maximum intensity (minimum SLP) and the location of cyclolysis. Additionally shown are the monthly mean baroclinicity (green contours; 25 to 45 by steps of $5\times10^{-6}$s$^{-1}$). The latitude of the maximum deepening is indicated by a black dashed contour.

the mean location at maximum deepening (second black dot along red line) is in all months either exactly (November, March) or very close to the location of the maximum in monthly mean baroclinic conversion (black contours in Fig. 7). This underlines that the selected tracks play an important role in shaping the monthly mean baroclinic conversion. The mean location of the maximum deepening is located farthest equatorward during January (Fig. 7b). We split these tracks into time steps before and

after the cyclones have crossed the latitude of the maximum deepening (dashed lines in Fig. 7) and compute box-and-whisker plots of the corresponding baroclinic conversion rates. The results show that in every month, baroclinic conversion rates are higher before the maximum deepening is reached. When comparing the three months, we find that baroclinic conversion is highest during January (black boxes in Fig. 8). An important feature of the suppression is the drop in baroclinic conversion in January after maximum deepening (difference between black and gray boxes in Fig. 8), which is larger in January compared to

November and March. Consequently, we arrive at similar conclusions as in the previous section. During January, the cyclones benefit from the increased baroclinicity early during the life cycle but due to the equatorward shift of the baroclinic zone the maximum in baroclinic conversion occurs at lower latitudes (second black dot along the red mean track in Fig. 7). The cyclones leave the baroclinic zone early on their way poleward, which can be deduced from the mean cyclone tracks in Fig. 7, and baroclinic conversion reduces to levels below that found during November and March after the cyclones crosses the latitude

where maximum deepening is observed.

## 7  Conclusions

This study presents a systematic analysis of the characteristics of cyclone life cycles over the North Pacific, with a particular focus on surface cyclones that propagate through the western North Pacific, where EKE decreases during midwinter (referred to as northern target region shown in Fig. 1). The goal of this study is to enrich the existing literature on the midwinter suppression of the North Pacific storm track with a systematic surface cyclone life cycle perspective to understand how cyclone life cycles change during midwinter in the western North Pacific.

The surface cyclone tracks feeding the storm track in the western North Pacific originate from three preferred regions: (i) downstream of Kamchatka, (ii) over the Kuroshio extension and (iii) over the East China Sea (Fig. 2). Kuroshio and Kamchatka cyclones dominate the total cyclone number over the western North Pacific throughout the cold season, while East China Sea cyclones become relevant during spring. Kamchatka and Kuroshio cyclones are preferentially triggered by upper-level waves entering the Pacific through the northern seeding branch, while East China Sea cyclones are at genesis low-level features (Chang, 2005). The analyzed tracks have their lysis mostly poleward of their genesis location in the mid-Pacific. Cyclones in the eastern North Pacific thus require requires further analysis.

Our key findings can be summarized as follows. The equatorward movement of the baroclinic zone in midwinter affects the life cycles of cyclones from all three genesis regions, but in a different way:

- Kamchatka cyclones develop in midwinter in a region of reduced baroclinicity. Compared to November, their lifetime decreases, the time to maximum deepening since genesis reduces and they become less intense. They contribute by about 40 % to the total cyclone number in winter over the western North Pacific, where EKE is suppressed. The weakening of Kamchatka cyclones is thus a crucial contribution to the suppression. Interestingly, despite the reduced baroclinicity in January, the number of Kamchatka cyclones is not reduced in midwinter. Kamchatka cyclones do not re-intensify during March, thus they do not benefit from the poleward movement of the baroclinic zone in spring.

- East China Sea cyclones benefit from the equatorward movement of the baroclinic zone in midwinter. Compared to November, they become more intense, the fraction of explosively deepening cyclones increases and their lifetime increases. They become weaker in March, but the fraction of explosively deepening life cycles remains higher than for Kuroshio and Kamchatka cyclones. In addition, their lifetime is longer in March compared to January. In March, East China Sea cyclones contribute by nearly 22% to the total cyclone number over the northwestern Pacific, while in fall and winter their contribution is approximately 15%. Thus, they seem to play a role in the re-intensification of the storm track during spring.

- The changes in the life cycles of Kuroshio cyclones are the most complex, but understanding these changes is crucial, because Kuroshio cyclones contribute strongest to the total cyclone number in the northern target region in midwinter (45%). Compared to the shoulder months, in January the lifetime of Kuroshio cyclones and the time to maximum deepening are shortest. The fraction of cyclones satisfying the "bomb cyclogenesis" criterion (Sanders and Gyakum, 1980) first reduces from November to January but then remains at similar levels until March. However, highest values

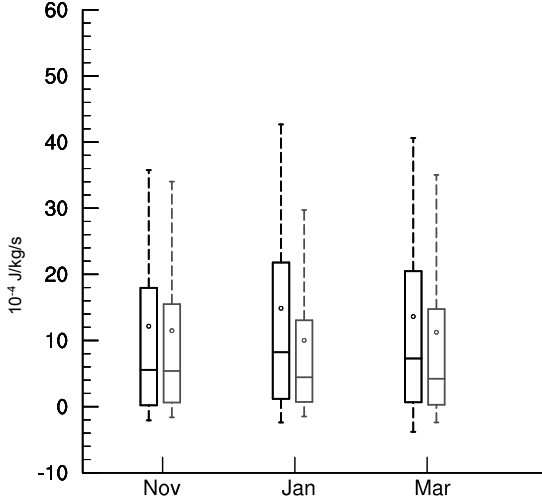

**Figure 8.** Box-and-whisker diagram for baroclinic conversion at $500\,\text{hPa}$ ($10^{-4}\,\text{J}\,\text{kg}^{-1}\,\text{s}^{-1}$) averaged within a radius of $1000\,\text{km}$ around the surface cyclone centers that enter the target region in Fig. 7 from upstream, before (black) and after (gray) passing the latitude of maximum deepening (black dashed line in Fig. 7).

in 6-hourly baroclinic conversion rates are found in January, but these occur at lower latitudes and they are sustained
for a reduced number of time steps relative to the shoulder months. In terms of minimum sea level pressure, Kuroshio cyclones are however most intense in January.

Overall, during midwinter, the life cycle of a Kuroshio cyclone is best characterized by a short and intense early deepening, in agreement with the higher baroclinicity, followed by a fast decay and poleward propagation away from the more equatorward located baroclinic zone. According to this interpretation, we observe an acceleration of the Kuroshio life cycle during mid-
winter. This interpretation is in agreement with the idea of a reduced baroclinic conversion efficiency because the efficiency is dictated by the vertical tilt of a cyclone (Schemm and Rivière, 2019). Acceleration of the life cycle with intense early growth results in cyclones that acquire a rather inefficient vertical tilt earlier in the life cycle. Kuroshio cyclones are thus in different months in different stages of their life cycle at similar latitudes. The stronger but earlier deepening followed by an earlier decay is the cyclone life-cycle perspective on the midwinter suppression over the western North Pacific.
It is also important to mention a few caveats of this study. Our results are based on a single object-based cyclone detection scheme. It is known that cyclone tracks are sensitive to the identification and tracking scheme (Neu et al., 2013), this holds also true for the genesis location. While they typically agree on deep systems, a higher sensitivity must be expected for shallow systems, as is the case for Kamchatka cyclones in January. Further, we ignore short-lived systems with a lifetime of less than 24 hours. Such systems might become more frequent in midwinter because the lifetime of all systems is reduced in January.
Baroclinic conversion occurs also in the absence of surface cyclones, for example by the propagation of an upper-level trough,

like the second baroclinic conversion peak in Fig. 4. Baroclinic conversion during non-cyclone days is also affected by the midwinter suppression and this reduction is not explained by our study. Our study has its focus on the western North Pacific, where climatological mean baroclinicity is highest and the reduction in baroclinic conversion and EKE is thus most surprising. Cyclone tracks that feed the eastern North Pacific are generated over the central-east Pacific (Hoskins and Hodges, 2002; Wernli and Schwierz, 2006) where climatologically barocliniciy is much lower compared with the western Pacific. Cyclone tracks over the eastern North Pacific not surprisingly have a large fraction of secondary cyclones (Fig. 5b in Schemm et al., 2018). Mechanisms responsible for the suppression in the eastern North Pacific thus require further analysis.

*Author contributions.* All three authors contributed to the discussion and final interpretation of the results. HB and SeS performed the analyses. SeS wrote, supported by HB and HW, the publication.

*Competing interests.* The authors declare no competing interests.

*Acknowledgements.* The authors acknowledge the discussions on the midwinter suppression with various colleagues at the 2018 storm track workshop on Utö, Sweden, and thank two anonymous reviewers for their constructive comments that helped to improve this paper. SeS has received funding from the European Research Council (ERC) under the European Union's Horizon 2020 research and innovation programme (grant agreement No. 848698). HB acknowledges funding from the Swiss National Science Foundation (SNSF grant 185049). We thank MeteoSwiss and ECMWF for access to the ERA-Interim reanalyses.

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
