# Peer review of "The Storm-Track Suppression over the Western North Pacific From a Cyclone Life-Cycle Perspective"

_Weather and Climate Dynamics, 2020_

## Referee Comment (RC1) · Anonymous Referee #1 · 21 Sep 2020

This is an interesting and well-written paper combining Eulerian and feature-tracking diagnostics to investigate midwinter suppression of the Pacific stormtrack. The topic and approach fit very well into the scope of WCD. I think that the conclusions and interpretation are well supported by the evidence presented and the manuscript does not suffer from flaws requiring major revision. However I have one comment that may require some minor additional analysis, and several requests for clarification and minor rewording.

Main comment:

The paper exclusively focuses on the western Pacific, and does a good job of account-

ing for midwinter suppression in that region. However, that region covers less than half of the area in the Pacific basin where suppression is observed to occur, which stretches eastward all the way to N America (Fig 1b). The authors note that their focus region is "located at the entrance of the storm track" (l.126), implying that eddies in that region will subsequently move downstream, so that the eastern part of the storm track will behave similarly to the western part. The implicit message is that a theory for suppression in the western region will also explain suppression in the Pacific storm track as a whole. But is this really true? After all, cyclones have a marked bias to poleward propagation, and it's not obvious they will follow the purely zonal propagation required by this implicit statement.

I think that leaving the reader guessing about this point risks being misleading, and requires clarification. For example, the authors could use the cyclone track data to show that cyclones passing through the northwestern "suppressed" box do indeed go on to feed the eastern part of the stormtrack where suppression is observed. Alternatively, they could omit further analysis, but provide a clear statement (in the abstract and conclusions) that mechanisms responsible for suppression in the east require further analysis.

Minor comments:

l. 46: "Subtropical jet regime": For the reader not deeply versed in the current literature, it would be useful to give a brief explanation of what you exactly mean by this expression (and what other regimes are possible).

l. 66: "propagate in tandem poleward": Fig 21 in Hoskins et al 1985 and surrounding text do not actually say anything about preferential poleward propagation, so far as I can see; the poleward propagation mechanisms instead are discussed in later work for example by Gwendal Riviere and Talia Tamarin, and possibly others I'm not familiar with. Some citations to literature on poleward propagation should be inserted here. This is clearly also relevant to my main comment above.

[Figure]

l. 110: please state the cutoff frequency used for the high-pass filtering.

l. 117: The analysis of EKE and baroclinic conversion in this and later sections is all carried out at 500 hPa. This choice needs some justification. Would analysis at other levels, or in the vertical average, give the same qualitative results and conclusions?

Fig 1: It would be useful to show a plot of cyclone track densities overlayed on EKE to appreciate their relationship (this could be done directly in Fig 1, or separately in supplementary material to avoid clutter)

l. 163, Table 1: please specify what exact genesis regions are used to define Kamchatka, Kuroshio and East China Sea cyclones.

l. 216 and elsewhere: I recommend sticking to the expression "feature tracking" or "cyclone tracking", rather than the vague and potentially misleading "quasi-Lagrangian". Many studies (including some by these authors) combine true Lagrangian analysis with feature tracking, in which case the inappropriateness of "quasi-Lagrangian" becomes obvious. Better for the community to have a single word for a single concept.

l. 245: Surface cyclones do not necessarily correspond only to deep (troposphere filling) eddies; they could also be shallow, diabatically maintained eddies. Some rewording may be needed here.

l. 265: Some quantification would be useful here: what fraction do cyclone days/non-cyclone days cumulatively contribute to mean baroclinic conversion, and to the suppression in January?

lines 292 and 301: Seems to me, by eye from Fig 6, that mean baroclinicity is reduced from Nov to Jan by about the same amount for both Kuroshio and Kamchatka cyclones. It's possible I'm misunderstanding here, in which case please clarify this point.

---

## Referee Comment (RC2) · Anonymous Referee #2 · 21 Sep 2020

This paper examines from a Lagrangian storm-following perspective, the theory that the Pacific midwinter suppression results from the baroclinic conversion becoming less efficient as the jet shifts equatorward to a more subtropical position during mid winter. In particular, it examines how this picture is modified by the fact that there are three different regions from which storms originate and seed the Pacific storm track. For that the authors track surface lows that reach two target regions, defined based on the changes in EKE between Nov and Jan (and between Jan and Mar), in the Eastern Pacific (the storm track entrance region). They then examine statistics of the evolution of the cyclones which pass through the two regions. I think the results are interesting, convincingly robust, and relevant, thus the work merits publication however there are a

few points which need addressing prior to publication.

After reading the paper and thinking of the results I am wondering why the authors did not define a single EKE target region, which moves from month to month with the EKE maximum, and performed the analysis this way, i.e. examining the storms which reach each month's region, separated to the different cyclogenesis regions. This would reduce confusion between a reduction of EKE due to a shifting relative to the averaging domain and a real overall reduction of the total storm energy. The main hesitation I have with the approach taken here is the fact that the two regions span around 15 and 10 degrees latitude- order of 1000-1500 km, which is on the order of typical cyclone radii. Thus I am guessing a cyclone will feel parts of both regions as it evolves and propagates along its track. The interpretation of a latitudinal shift in terms of a dipole is less intuitive on a single storm scale. It sounds intuitive reading the paper since the authors discuss tracks that pass through each region but that in some sense gives a wrong picture. I am not saying the approach is wrong but the authors should somehow justify it, at the very least by a discussion of spatial scales, why they choose to divide the domain this way, and how the results relate to the physical picture of single cyclones. Best will be of course to compare the analysis for single regions which shift with the EKE maximum.

Also, it is not clear at the moment if the main contribution of the paper is in elucidating the changes in the eddies which contribute to the midwinter suppression and the dependence on the cyclogenesis region, or if it provides a more fundamental understanding by further by also explaining the changes in the eddies. For the latter, the authors need to tighten the discussion of how the results fit in with existing theory.

There are a few confusing points in the discussion, which I will try to point out here:

- The main underlying theory - that equatorward shifting of the jet results in a weakening of the storms due to their meridional tilt, inherently looks at the entire storm and how its meridional shift varies with height - the division into poleward and equatorward parts in

this argument does not necessarily make sense.

- The argument that the baroclinicity shifts equatorwards into the Kuroshio cyclogenesis region during mid winter, suggests at first that the storms should grow more efficiently during mid winter, but the overall argument made is that they grow less efficiently. I think the answer to this is given in the summarizing argument, on lines 338-345, but I am not sure I fully understand it- do the authors mean to say that the larger meridional tilt seen in Schemm and Riviere is in a sense an artifact of the time averaging over the cyclone life cycle, and since the cyclone moves poleward quicker, while undergoing faster growth and decay as it shifts poleward, the time averaged structure has a stronger tilt? Thus the overall growth over the full cyclone life cycle is what becomes less efficient? This in essence sounds similar to the original arguments by Nakamura (1992), that storms grow faster but also move quicker, but instead of the stronger zonal wind advecting the storms out of the baroclinicity region, the storms move poleward and they undergo the full nonlinear life cycle of growth and decay..

- Schemm and Riviere discuss Nakamura and Sampe's argument that the growth is less efficient on a strong and subtropical jet due to a stronger meridional tilt which the storms assume if their surface cyclogenesis remains at the same latitude. They point out that the meridional-vertical tilt implied by Nakamura's argument (equatorwards with height) is opposite to the tilt they find (poleward with height). They mention that the meridional tilt would be different for different seeding latitudes (I assume this is part of the motivation for this paper). I think the authors should more explicitly tie the current results to this argument, and specifically does the change found in Kamachatka cyclone life cycles fit with the argument of Nakamura and Sampe?

- The main results for the Kamachatka cyclones (lines 334-337): "The fraction of explosively deepening cyclones first reduces from November to January but then 335 remains at similar levels until March. Highest values in baroclinic conversion are found during midwinter, but these occur at lower latitudes, south of the northern target region, and they are sustained for a reduced number of time steps. In terms of minimum sea

level pressure, Kuroshio cyclones are most intense in January." The finding of a reduction in explosive cyclogenesis but more intense cyclones during January is confusing. Also- is it obvious why the growth in mid winter is sustained for less time?

Specific comments:

Figure 2: What is counted as propagation through a region- that the cyclone track which follows the cyclone center (a single pixel of minimum pressure?) pass through it, or a part of the cyclone (the region of 1's corresponding to the detection scheme) passes through it? Similarly- the cyclogenesis is counted as the whole cyclone or its center?

Figure 3: I am not sure I understand what is shown here - the caption says "relative contributions...to the total surface cyclone frequency in the northern target region", which implies a very wide cyclogenesis region to the west and north of the target region, which is not what I expect, and I am not sure how this fits with figure 2..? The plots look more like the contribution to total cyclone frequency from those cyclones originating in the target area. But then the percentage is out of the total cyclones contributing to the target region, but not including cyclones which miss the target region? so the sum of the right and left columns add to 100% in the target region but not outside of it? An explicit explanation of how the fields in figure 3 relate to those in figure 2 might help clear things.

Do you have any idea why the number of Kamachatka cyclones decreases and the number of East china sea cyclones increase as the season progresses?

Section 2: Methodology - using a monthly mean static stability alongside low and high pass filtered quantities - how do you deal with the jumps in static stability in between months? how much does the static stability change from month to month? Do you use the climatology or each year's monthly mean?

The discussion on page 7 needs some tightening - there is repetition of the results of

the previous sections and within the section itself.

line 265- Please state explicitly why you say the non cyclone days contribute *much* more than non cyclone days- they clearly contribute more but its not clear on quick look that its all that much more. Being more quantitative might help.

line 278- the authors average at a radius of 1000km around the cyclone center. 1000km is roughly the latitudinal length of the southern box, so if the cyclone is at the southern edge of the EKE decrease box, the averaging could include a very large portion of the EKE increase region as well. . . is this problematic and how does this affect the results? see major comment above.
* * *

---

## Editor Comment (EC1) · David Battisti (Editor) · 1 Oct 2020

The two anonymous reviewers are scientists who are experts in the relationship between synoptic dynamics and climate. Their reviews are exceptionally thorough and insightful; addressing all of their major concerns will improve the manuscript.

Please pay particular attention to the following concerns, which I also flagged in my reading of the manuscript:

- Reviewer 2 is concern about how the target region is defined, and I agree. Presently the two boxes used in the analysis do well to describe why the maximum EKE shifts equatorward in going from November to January, but it is not clear how or why that relates to the mid-winter minimum in maximum EKE. In all three months analyzed, the maximum in EKE is found at about 43N and it straddles the two boxes and so the boxes don't capture the traditional view of the midwinter minimum in EKE: a reduction in the *maximum* of EKE. How would the results and conclusions differ if one box was used that was centered on the maximum EKE?

- Both reviewers note that it isn't clear that changes in the baroclinic conversion on cyclone days is that much greater than on non-cyclone days, and I agree. As per reviewer 1's request, some quantification of this statement is required. Reading the mean values off Fig. 5, the relative daily contributions to drop in baroclinic conversion between Nov and Jan on non-surface cyclone days is (~13 J/kg/s *.55 – 3.5 J/kg/s *.51)*M = 5.4 M J/kg/s (where M is the number of seconds in a day), which is as large as that due to the drop due to surface cyclones days (~17 *.45 – 7*.49)*M = 4.2 M J/kg/s. And that raises an important concern: if the conversion on 'non-surface cyclone days' is due to upper level cyclones, this suggest the upper level cyclone changes contribute as much as surface cyclones to the change in total EKE. It seems that the reduction in baroclinic conversion that is not associated with a surface level cyclone is as important to the change in EKE (and perhaps in the midwinter minimum in the maximum EKE) as is the change in vertical structure of cyclones with a surface signature.

- Reviewer 1 is concerned that the analysis only addresses the changes in EKE in the far western Pacific, and that no evidence is presented for the changes in the central and eastern Pacific. Review 1 suggests either the authors perform a relatively straightforward calculation to demonstrate the relevance of the far western Pacific results to the bulk of the Pacific, or be clear that the conclusions are specific to mechanisms for the changes in EKE

in the far western Pacific and future analysis should be done to address changes in the central and eastern Pacific. I will leave it to you to choose which way to go here.

A separate question I have (related to a point raised by Reviewer 2) concerns the use of a monthly mean of stability S in the calculation of baroclinic conversion. There is a lot of low frequency variability associated with the stationary wave coming off east Asia: how would the results differ if a low passed version of S was used – the same filter used to estimate theta_bar in this calculation?

---

## Author Response (AR1)

Reply to the editor's comments

We would like to thank the editor for concisely summarizing the concerns of both reviewers and sharing further comments and ideas that have helped us to further improve the manuscript.

**Editor comment #1:** How would the results and conclusions differ if one box was used that was centered on the maximum EKE?
Reply: This is a good idea. A detailed analysis of a box shifting with the maximum in baroclinic conversion and further comments can be found in our reply to review #2.

**Editor comment #2:** Both reviewers note that it isn't clear that changes in the baroclinic conversion on cyclone days is that much greater than on non-cyclone days, and I agree. As per reviewer 1's request, some quantification of this statement is required. It seems that the reduction in baroclinic conversion that is not associated with a surface level cyclone is as important to the change in EKE (and perhaps in the midwinter minimum in the maximum EKE) as is the change in vertical structure of cyclones with a surface signature.
Reply: Yes, this is correct, the relative changes are similar. A quantification is given in the revised manuscript and also in the reply to review #1. Absolute values of baroclinic conversion are larger during cyclone days, but the relative change in the conversion on non-cyclone days from November to January is quite similar to that of cyclone days. Note, the definition of non-cyclone days allows that the target region is covered by up to 25% by a surface cyclone. We do agree that baroclinic conversion associated with upper level eddies that are shallow (not extending through the depth of the troposphere) is also suppressed.

**Editor comment #3:** Reviewer 1 is concerned that the analysis only addresses the changes in EKE in the far western Pacific, and that no evidence is presented for the changes in the central and eastern Pacific.
Reply: We fully agree. We now provide clear statements in the abstract and conclusions. The eastern Pacific requires further investigation. We also adapted the title of our study. For more details see the reply to reviewer #1.

**Editor comment #4:** A separate question I have (related to a point raised by Reviewer 2) concerns the use of a monthly mean of stability S in the calculation of baroclinic conversion. There is a lot of low frequency variability associated with the stationary wave coming off east Asia: how would the results differ if a low passed version of S was used –the same filter used to estimate theta_bar in this calculation?
Reply: During the preparation of the data for Schemm and Rivière (2019), which uses the same data set, we tested different ways to compute S and the results only marginally differed. We thus decided to stick to the traditional way of defining S based on a vertical reference temperature profile of the monthly mean as in Cai and Mak (1990) or Orlanski and Katzefy (1991).

Reply to the reviewer's comments

Anonymous Referee #1

We would like to thank both reviewers for carefully evaluating our manuscript and for providing comments that helped us to further improve our study.

**Reviewer:** The paper exclusively focuses on the western Pacific, and does a good job of accounting for midwinter suppression in that region. However, that region covers less than half of the area in the Pacific basin where suppression is observed to occur, which stretches eastward all the way to N America (Fig 1b). The authors note that their focus region is "located at the entrance of the storm track" (l.126), implying that eddies in that region will subsequently move downstream, so that the eastern part of the storm track will behave similarly to the western part. The implicit message is that a theory for suppression in the western region will also explain suppression in the Pacific storm track as a whole. But is this really true? After all, cyclones have a marked bias to poleward propagation, and it's not obvious they will follow the purely zonal propagation required by this implicit statement.
I think that leaving the reader guessing about this point risks being misleading, and requires clarification. For example, the authors could use the cyclone track data to show that cyclones passing through the northwestern "suppressed" box do indeed go on to feed the eastern part of the stormtrack where suppression is observed. Alternatively, they could omit further analysis, but provide a clear statement (in the abstract and conclusions) that mechanisms responsible for suppression in the east require further analysis.

**Authors:** This is an excellent point. We focus on the part of the storm track over the western North Pacific where baroclinicity is largest in midwinter. The cyclone tracks analyzed in our study have their lysis on average poleward of 50° N in the central Pacific. The Pacific storm track is known to "restart" over the central Pacific. Hoskins and Hodges (2002; p. 1060) noted "*that very few synoptic systems can be tracked along the length of the Pacific storm track. Indeed, most of the systems generated over eastern Asia do not even reach the mid-Pacific. It is the systems that are generated in the central-east Pacific that occlude on the northwest coast of North America.*" A finding, which was later confirmed by Wernli and Schwierz (2006; Figs. 9c and 9d). Indeed, a large fraction of the cyclogenesis over the eastern Pacific is secondary cyclogenesis (Schemm et al. 2018; Fig. 5b). We therefore fully agree with the reviewer that our study, focusing on the western North Pacific, does not address midwinter suppression of the entire Pacific storm track. To potentially explain the suppression over the eastern Pacific, it could be rewarding to explore suppression of the downstream development (Simmons and Hoskins 1979, Orlanski and Chang 1993), which would be of high scientific merit and should be reserved for a follow-up study.
In the revised manuscript, we adapted the title and we provide a clear statement in the abstract and the conclusions that we focus exclusively on the western North Pacific where climatological mean baroclinicity is highest and that the suppression over the eastern Pacific requires additional analysis.

**Minor comments**

- l. 46: "Subtropical jet regime": For the reader not deeply versed in the current literature, it would be useful to give a brief explanation of what you exactly mean by this expression (and what other regimes are possible).
  Authors: We added an explanation.

- l. 66: "propagate in tandem poleward": Fig 21 in Hoskins et al 1985 and surrounding text do not actually say anything about preferential poleward propagation, so far as I can see; the poleward propagation mechanisms instead are discussed in later work for example by Gwendal Riviere and Talia Tamarin, and possibly others I'm not familiar with. Some citations to literature on poleward propagation should be inserted here. This is clearly also relevant to my main comment above.
  Authors: Yes, we added more appropriate references. It is long-standing knowledge from case studies (Palmén and Newton 1969), also discussed in idealized experiments (Hoskins and West 1979, Davies et al. 1991) and later from feature-based climatologies (e.g., Hoskins and Hodges 2002) that cyclone tracks are deflected poleward. Important recent studies about the underlying mechanism are by Gilet et al. (2009), Rivière et al. (2012) and Tamarin and Kaspi (2017). In our study, the poleward motion can be inferred from Fig. 3, which shows that less than 20% of all cyclone tracks in the Gulf of Alaska are generated over the Kuroshio.

- l. 110: please state the cutoff frequency used for the high-pass filtering.
  Authors: We added this information. It is 10 days.

- l. 117: The analysis of EKE and baroclinic conversion in this and later sections is all carried out at 500 hPa. This choice needs some justification. Would analysis at other levels, or in the vertical average, give the same qualitative results and conclusions?
  Authors: The level is a pragmatic choice. The midwinter increases with altitude (Schemm and Schneider 2018), but baroclinic conversion is typically largest in the lower troposphere. At the 500 hPa level, baroclinic conversion is still large and the midwinter suppression is a well-marked feature in the conversion rates. Qualitatively, we expect the same results for vertical averages or integrals. For example, the study by Schemm and Schneider (2018) was based on vertically integrated conversion rates and yielded comparable results. The location of our target regions is also not affected by the choice of the level and all results related to the surface cycle tracks are thus largely independent of the vertical level.

- Fig 1: It would be useful to show a plot of cyclone track densities overlayed on EKE to appreciate their relationship (this could be done directly in Fig 1, or separately in supplementary material to avoid clutter).
  Authors: We show cyclogenesis frequencies and considered adding the surface cyclone frequencies to Fig. 1 and Fig. 3., but the cyclone frequency peaks poleward of the EKE maximum and provide no new insight into the nature of the suppression. A figure showing EKE and surface cyclone frequencies overlayed is Fig. 1 in Schemm and Schneider (2018). Following the reviewer's suggestion, we show the mean position of the tracks (Fig. R2 in reply document to Reviewer #2) and we will add it to the final version our study.

- l. 163, Table 1: please specify what exact genesis regions are used to define Kamchatka, Kuroshio and East China Sea cyclones.
  Authors: We added this information to the table.

- l. 216 and elsewhere: I recommend sticking to the expression "feature tracking" or "cyclone tracking", rather than the vague and potentially misleading "quasi-Lagrangian". Many studies (including some by these authors) combine true Lagrangian analysis with feature tracking, in which case the inappropriateness of "quasi-Lagrangian" becomes obvious. Better for the community to have a single word for a single concept.
  Authors: We fully agree and removed "quasi-Lagrangian" from the manuscript.

- l. 245: Surface cyclones do not necessarily correspond only to deep (troposphere filling) eddies; they could also be shallow, diabatically maintained eddies. Some rewording may be needed here.
  Authors: Correct, we adapted the sentence accordingly.

- l. 265: Some quantification would be useful here: what fraction do cyclone days/non-cyclone days cumulatively contribute to mean baroclinic conversion, and to the suppression in January?
  Authors: On cyclone days, the conversion reduces from $17 \times 10^4$ J/kg/s in November to $6.4 \times 10^4$ J/kg/s in January and on non-cyclone days from 13 to $3.4 \times 10^4$ J/kg/s in the same period. In both categories, we find that the suppression is approximately $10 \times 10^4$ J/kg/s and the number of days in both categories is about 50%. In the revised paper, we highlight this more prominently. We also include the exact numbers in the corresponding section. We also emphasize that a non-cyclone day might still be affected by a cyclone, which propagates in close proximity or along the edge of the target region. But it is now correctly mentioned that baroclinic conversion on non-cyclone days is equally affected by the suppression and the relative reduction is similar.

- lines 292 and 301: Seems to me, by eye from Fig 6, that mean baroclinicity is reduced from Nov to Jan by about the same amount for both Kuroshio and Kamchatka cyclones. It's possible I'm misunderstanding here, in which case please clarify this point.
  Authors: This is correct, mean baroclinicity reduces by the same amount for both cyclone categories from November to January. We added a new panel to Fig. 6, which shows the conversion efficiency. It is not only the reduction in mean baroclinicity that matters, but Kuroshio and Kamchatka cyclones also become less efficient in converting the mean baroclinicity into eddy total energy. The reduction in baroclinicity and conversion efficiency contribute both to the reduction in baroclinic conversion. For example, the mean baroclinicity over the northern target region is similar for Kuroshio and Kamchatka cyclones, however Kamchatka cyclones have lower conversion rates due to an overall lower conversion efficiency. But it is correct, the relative change from November to January in the mean baroclinicity and conversion efficiency is for both the same. We clarified our reasoning in this section.

Literature:

Davies, H. C., Schär, C., & Wernli, H. (1991). The palette of fronts and cyclones within a baroclinic wave development. Journal of the atmospheric sciences, 48(14), 1666-1689.

Gilet, J. B., Plu, M., & Rivière, G. (2009). Nonlinear baroclinic dynamics of surface cyclones crossing a zonal jet. Journal of the atmospheric sciences, 66(10), 3021-3041.

Hoskins, B. J., & Hodges, K. I. (2002). New perspectives on the Northern Hemisphere winter storm tracks. Journal of the Atmospheric Sciences, 59(6), 1041-1061.

Orlanski, I., & Chang, E. K. (1993). Ageostrophic geopotential fluxes in downstream and upstream development of baroclinic waves. Journal of the atmospheric sciences, 50(2), 212-225.

Palmén, E. H., & Newton, C. W. (1969). Atmospheric circulation systems: their structure and physical interpretation (Vol. 13). Academic press.

Rivière, G., Arbogast, P., Lapeyre, G., & Maynard, K. (2012). A potential vorticity perspective on the motion of a mid-latitude winter storm. Geophysical research letters, 39(12).

Schemm, S., Sprenger, M., & Wernli, H. (2018). When during their life cycle are extratropical cyclones attended by fronts?. Bulletin of the American Meteorological Society, 99(1), 149-165.

Schemm, S., & Schneider, T. (2018). Eddy lifetime, number, and diffusivity and the suppression of eddy kinetic energy in midwinter. Journal of Climate, 31(14), 5649-5665.

Simmons, A. J., & Hoskins, B. J. (1979). The downstream and upstream development of unstable baroclinic waves. Journal of the Atmospheric Sciences, 36(7), 1239-1254.

Tamarin, T., & Kaspi, Y. (2017). Mechanisms controlling the downstream poleward deflection of midlatitude storm tracks. Journal of the Atmospheric Sciences, 74(2), 553-572.

Wernli, H., & Schwierz, C. (2006). Surface cyclones in the ERA-40 dataset (1958–2001). Part I: Novel identification method and global climatology. Journal of the atmospheric sciences, 63(10), 2486-2507.

Reply to the reviewer's comments

Anonymous Referee #2

We would like to thank both reviewers for carefully evaluating our manuscript and for providing comments that helped us to further improve our study.

**Reviewer:** After reading the paper and thinking of the results I am wondering why the authors did not define a single EKE target region, which moves from month to month with the EKE maximum, and performed the analysis this way, i.e. examining the storms which reach each month's region, separated to the different cyclogenesis regions. This would reduce confusion between a reduction of EKE due to a shifting relative to the averaging domain and a real overall reduction of the total storm energy. The main hesitation I have with the approach taken here is the fact that the two regions span around 15- and 10-degrees latitude-order of 1000-1500 km, which is on the order of typical cyclone radii. Thus, I am guessing a cyclone will feel parts of both regions as it evolves and propagates along its track. The interpretation of a latitudinal shift in terms of a dipole is less intuitive on a single storm scale. It sounds intuitive reading the paper since the authors discuss tracks that pass through each region but that in some sense gives a wrong picture. I am not saying the approach is wrong but the authors should somehow justify it, at the very least by a discussion of spatial scales, why they choose to divide the domain this way, and how the results relate to the physical picture of single cyclones. Best will be of course to compare the analysis for single regions which shift with the EKE maximum.

**Authors:** We very much thank the reviewer for these thoughtful comments and we appreciate the suggestion of choosing a target box that shifts with the EKE maximum.

During January and March, the EKE maximum at the 500 hPa level is located over the mid-Pacific, but in November over the eastern Pacific (black contours in Fig. R1). The corresponding cyclone tracks are generated in very different environments (cyclogenesis frequencies are shown in Fig. R1). Hoskins and Hodges (2002; p. 1060) noted *"that very few synoptic systems can be tracked along the length of the Pacific storm track. Indeed, most of the systems generated over eastern Asia do not even reach the mid-Pacific. It is the systems that are generated in the central-east Pacific that occlude on the northwest coast of North America."* We argue that comparing cyclone tracks generated in such different environments is potentially confusing. Instead, in agreement with Reviewer #1, we propose to focus on the western North Pacific where the maximum in baroclinic conversion occurs throughout the winter. Climatologically, mean baroclinicity is highest over the western Pacific and it increases from November to January (green contours in Fig. R2). Similar to the suppression of EKE, there is, however, a reduction of the baroclinic conversion during this time period (black contours in Fig. R2), which is "unexpected" given the increase in baroclinicity. We therefore propose a modification of the suggestion by the referee and choose a target region that shifts with the maximum in baroclinic conversion over the western Pacific.

[Figure]

*Figure R1: Cyclogenesis frequency (color shading; %) for surface cyclone tracks that propagate through a target region (shown as a gray box) that shifts with the EKE maximum for (left) November, (mid) January and (right) March. EKE at 500 hPa is shown by black contours.*

[Figure]

*Figure R2: Baroclinic conversion at 500 hPa (black contours; $10^4$ J kg-1 s-1) and change relative to the previous month (shading). Gray boxes denote a target region that shifts with the maximum in baroclinic conversion. Also shown is the mean position of cyclone tracks that enter the target region from upstream (red line), with black dots marking the mean position of cyclogenesis, maximum deepening (6-hourly SLP change), maximum intensity (minimum SLP) and cyclolysis.*

A target region that shifts with the maximum in baroclinic conversion is shown in Fig. R2. Also shown as a red line is the mean position of the cyclone tracks that propagate into this region from upstream, excluding tracks with genesis in the box (Fig. R1). The first black dot indicates the mean cyclogenesis location, the second the mean location of maximum deepening (6-hourly SLP change), the third the location of maximum intensity (minimum SLP), and the fourth indicates the mean cyclolysis location. During November and March, the location of maximum deepening (second dot) coincides with the maximum in monthly mean baroclinic conversion (black contours). During January, it is slightly north of it. We now repeat the statistical analysis of these tracks. To this end, we split all tracks in time steps before and after the tracks cross the latitude of the monthly mean maximum deepening (shown as the dashed line in Fig. R2) and we next analyze baroclinic conversion averaged in a 1000 km radius at every time step before and afterwards (similar as in the main manuscript).

The corresponding box-and-whisker plot of baroclinic conversion (Fig. R3) shows that baroclinic conversion is highest in January but only before the maximum deepening is reached (this is in agreement with highest baroclinicity in January at latitudes equatorward of the latitude of maximum conversion). Once the tracks have passed the location of maximum deepening, the baroclinic conversion is reduced relative to the time steps before maximum deepening (compare black to gray box-and-whiskers in every month in

[Figure]

*Figure R3: Box-and-whisker diagram for baroclinic conversion at 500 hPa (104 J kg-1 s-1) averaged within a radius of 1000 km around the surface cyclone centers of surface cyclone tracks that enter the target region shown in Fig. 2 from upstream for time steps south of (black) and north of (gray) the maximum deepening (6-h SLP change).*

Fig. R3) but baroclinic conversion is now also suppressed in January relative to November in March (compare gray box-and-whiskers between the different months). This is in agreement with the red mean cyclone track in Fig. R2, which on its way poleward propagates out of the contours of largest monthly mean baroclinic conversion in January (black contours in Fig. R2) but in November and Marc. Overall, the results nicely complement our results presented in our manuscript in Fig. 6, where we split the tracks in time steps inside and outside of a "northern target region". We will add the above new results as a new section to the revised manuscript and we thank the reviewer for suggesting a shifting target box.

**Reviewer:** Also, it is not clear at the moment if the main contribution of the paper is in elucidating the changes in the eddies which contribute to the midwinter suppression and the dependence on the cyclogenesis region, or if it provides a more fundamental understanding by further by also explaining the changes in the eddies. For the latter, the authors need to tighten the discussion of how the results fit in with existing theory.
**Authors:** The main goal of our study is to describe how the life cycle of cyclones generated in the different genesis region, which all affect the region of suppression in the western North Pacific, changes during midwinter.

**Reviewer:** The main underlying theory - that equatorward shifting of the jet results in a weakening of the storms due to their meridional tilt, inherently looks at the entire storm and how its meridional shift varies with height - the division into poleward and equatorward parts in this argument does not necessarily make sense.
**Authors:** We are undecided to which argument in our manuscript the reviewer is referring to. We separate the cyclone tracks into sections equatorward and poleward of a critical latitude (above in Fig. R2 the latitude of maximum baroclinic conversion) but unrelated to the vertical tilt.

**Reviewer:** The argument that the baroclinicity shifts equatorwards into the Kuroshio cyclogenesis region during midwinter, suggests at first that the storms should grow more efficiently during mid-winter, but the overall argument made is that they grow less efficiently. I think the answer to this is given in the summarizing argument, on lines 338-345, but I am not sure I fully understand it- do the authors mean to say that the larger meridional tilt seen in Schemm and Riviere is in a sense an artifact of the time averaging over the cyclone life cycle, and since the cyclone moves poleward quicker, while undergoing faster growth and decay as it shifts poleward, the time averaged structure has a stronger tilt? Thus, the overall growth over the full cyclone life cycle is what becomes less efficient? This in essence sounds similar to the original arguments by Nakamura (1992), that storms grow faster but also move quicker, but instead of the stronger zonal wind advecting the storms out of the baroclinicity region, the storms move poleward and they undergo the full nonlinear life cycle of growth and decay.
**Authors:** We added the baroclinic conversion efficiency to our analysis. It is shown in the revised Fig. 6 of the manuscript and shown below as Fig. R4 of this document. Similar as before, we split the Kuroshio tracks into time steps outside and inside of the northern target region. As suggested by the reviewer, the cyclones are more efficient in January compared with November and March as long as they are equatorward of the target region, but once they propagated poleward and outside the baroclinic zone they quickly become less efficient. This is in agreement with what is shown in brown and blue shading Fig. R4 in Schemm and Rivière (2019). We hope that this clarifies our reasoning: First the cyclones are more efficient and baroclinic conversion rates are large, but further poleward their efficiency is reduced. We do

[Figure]

Figure R4: (revised version of Fig. 6 in the manuscript): Box-and-whisker diagram for (left) baroclinic conversion at 500 hPa (10-4 J kg-1s-1) averaged within a radius of 1000 km around the surface cyclone centers of Kuroshio and East China Sea cyclones, before (black) and after (gray) entering the northern target region, and for Kamchatka cyclones (blue). Additionally, shown are the mean background baroclinicity along the tracks (red horizontal lines) and the percentage of time steps before and after entering the target region. (right) Similar as in the left panel but for the baroclinic conversion efficiency (unitless).

not estimate the propagation speed nor do we perform any time averaging along the tracks, but we do agree that in essence the argumentation is similar to Nakamura (1992) that storms move quicker out of the baroclinicity region, but they also become less efficient on their way poleward. We add this link to the original 1992 paper to the corresponding section.

Reviewer: Schemm and Riviere discuss Nakamura and Sampe's argument that the growth is less efficient on a strong and subtropical jet due to a stronger meridional tilt which the storms assume if their surface cyclogenesis remains at the same latitude. They point out that the meridional-vertical tilt implied by Nakamura's argument (equatorwards with height) is opposite to the tilt they find (poleward with height). They mention that the meridional tilt would be different for different seeding latitudes (I assume this is part of the motivation for this paper). I think the authors should more explicitly tie the current results to this argument, and specifically does the change found in Kamchatka cyclone life cycles fit with the argument of Nakamura and Sampe?

Authors: We try to tie our work better to the previous literature using Fig. R4 to which we added the baroclinic conversion efficiency. The figure shows that the cyclones become less efficient when propagating poleward. The main motivation of this paper is the life cycle perspective: How does the conversion and the efficiency change during the life cycle and which cyclones contribute to the reduction in the baroclinic conversion? In Schemm and Rivière (2019) we speculate that the argument of Nakamuara and Sampe applies to the southern seeding branch. According to Chang (2005), the southern seeding branch is associated with East China Sea cyclones (though the upper level waves do not necessarily trigger East China Sea cyclones) and can also sometimes trigger Kuroshio cyclones (though they are mostly triggered by the northern seeding branch). To connect the individual cyclone tracks to one of the two seeding branches would be beyond the scope of this study.

Reviewer: The main results for the Kamchatka cyclones (lines 334-337): "The fraction of explosively deepening cyclones first reduces from November to January but then remains at similar levels until March. Highest values in baroclinic conversion are found during midwinter, but these occur at lower latitudes, south of the northern target region, and they are sustained for a reduced number of time steps. In terms of minimum sea level pressure, Kuroshio cyclones are most intense in January." The finding of a reduction in explosive cyclogenesis but more intense cyclones during January is confusing. Also- is it obvious why the growth in mid-winter is sustained for less time?

**Authors**: We tried to clarify this point. The fraction of "bomb" cyclones reduces (deepening larger than 24 hPa in 24 hours normalized to 60° N as in Sanders and Gyakum 1980) from November to January, but the largest 6-hourly baroclinic conversion rates are still observed during January. They are maintained for a shorter time period, which matters most for the overall deepening. We do not know if this is obvious, but the life time of cyclones reduces in midwinter and the number of time steps with high growth rates is lowered.

**Reviewer:** Figure 2: What is counted as propagation through a region- that the cyclone track which follows the cyclone center (a single pixel of minimum pressure?) pass through it, or a part of the cyclone (the region of 1's corresponding to the detection scheme) passes through it? Similarly- the cyclogenesis is counted as the whole cyclone or its center?

**Authors:** For the selection of the tracks we use the minimum SLP as the center of the cyclone and the cyclone center must be at least once within the target region. We added this information to the manuscript. For the cyclogenesis, we compute a radius of 500 km around the genesis location to define a "genesis region". In this way, all genesis events are treated equally.

**Reviewer:** Figure 3: I am not sure I understand what is shown here - the caption says "relative contributions...to the total surface cyclone frequency in the northern target region", which implies a very wide cyclogenesis region to the west and north of the target region, which is not what I expect, and I am not sure how this fits with figure 2..? The plots look more like the contribution to total cyclone frequency from those cyclones originating in the target area. But then the percentage is out of the total cyclones contributing to the target region, but not including cyclones which miss the target region? so the sum of the right and left columns add to 100% in the target region but not outside of it? An explicit explanation of how the fields in figure 3 relate to those in figure 2 might help clear things.

**Authors:** The plot needs a better explanation. The plots are the contribution to the total cyclone frequencies propagating through the target area not originating from the target area. Additionally, we separate those that enter the target area from the south from all others, which essentially separated Kuroshio and East China Sea cyclones from Kamchatka cyclones. In the target region it adds up to 100% but not outside.

**Reviewer:** Do you have any idea why the number of Kamchatka cyclones decreases and the number of East china sea cyclones increase as the season progresses?

**Authors:** We do not have an explanation for the increase in East China Sea cyclones. For Kamchatka cyclones is seems to be the reduction in the mean baroclinicity and eventually also a reduction in the upper-level seeding, but this hypothesis would require further testing.

**Reviewer:** Section 2: Methodology - using a monthly mean static stability alongside low and high pass filtered quantities - how do you deal with the jumps in static stability in between months? how much does the static stability change from month to month? Do you use the climatology or each year's monthly mean?

**Authors:** During the preparation for the study of Schemm and Rivière (2019), we tested different filters and averaging windows and did not find a significant difference. For the static stability, the reference temperature profile is in agreement with the traditional literature computed from a climatological monthly mean.

**Reviewer:** The discussion on page 7 needs some tightening - there is repetition of the results of the previous sections and within the section itself.

**Authors:** Page 7 is the method section; we would appreciate if the reviewer could point us to the corresponding section that needs some tightening because we are unsure whether the reviewer refers to page 17 or section 7.

**Reviewer:** line 265- Please state explicitly why you say the non cyclone days contribute *much* more than non cyclone days- they clearly contribute more but its not clear on quick look that its all that much more. Being more quantitative might help.

Authors: In the revised version, we give the exact numbers of the change from cyclone days and non-cyclone days between November and January. We highlight the fact that cyclone days have much higher values in baroclinic conversion, but, as pointed out correctly by both reviewers, the changes are about the same (revised l. 275-280).

**Reviewer:** line 278- the authors average at a radius of 1000km around the cyclone center. 1000km is roughly the latitudinal length of the southern box, so if the cyclone is at the southern edge of the EKE decrease box, the averaging could include a very large portion of the EKE increase region as well. . . is this problematic and how does this affect the results? see major comment above.

Authors: Yes, this is correct, if the cyclone is near the southern edge of the northern target region it will tap into the southern target region. In Schemm and Rivière (2019) it was shown that when using a radius of 2000 km or an average inside the outermost closed SLP contour used for the tracking, the results are qualitatively still very similar to the 1000 km radius, though smaller (larger) in absolute values, respectively. We believe that the increase/decrease pattern in EKE is partly a result of the changes in baroclinic conversion along the tracks of the analyzed cyclones and when a cyclone affects first the southern box, for some time steps both target regions and further poleward only the northern target region then this is what results in the climatological mean and a transition zone does not seem to be problematic for the interpretation of our results.

[revised manuscript text omitted]

---

## Author Response (AR2)

**Reply to the reviewer's comments (Referee #2)**

We would like to thank the reviewer for carefully evaluating our revised manuscript.

**Reviewer:** The authors have improved the presentation and have added analyses and some discussions. Overall the paper is a nice and important contribution to our understanding of midwinter suppression specifically, and more generally, to how the statistics of single-storm evolutions combine to give the Eularian diagnostics of storm tracks. The authors added an analysis of eddy statistics in an area centered about the seasonally varying climatological baroclinic conversion, which is where the eddies draw energy from the mean flow APE. This is a nice choice since the peak baroclinic conversion region, which indicates where growth via extraction of mean PE, remains concentrated in the western Pacific. I agree with the authors that this is better than my initial suggestion to examine the statistics following the EKE maximum which peaks in the central pacific and is fed by cyclones originating in different regions than are shown in figure 2. I find the paper worthy of publication, but suggest making some minor clarifications.

**Authors:** We are pleased to read that our revisions have adequately addressed the reviewer's concerns. We would like to thank the reviewer for suggesting the analysis of a moving box, which nicely expanded our study.

**Reviewer:** The authors added a calculation of the baroclinic conversion efficiency, and this helps clarify some of the statements in the original version, however, I think the authors should add a paragraph in section 2.2 discussing what this quantity means qualitatively, and not just refer the reader to Schemm and Riviere's 2019 paper. The way I see it, after going through Schemm and Riviere (2019) is that the baroclinic conversion efficiency isolates the influence of the eddy structure (vertical-horizontal tilt of the geopotential height surfaces relative to the mean flow temperature gradients) on the magnitude of the baroclinic conversion, as opposed to the influence of the mean flow baroclinicity.

**Authors:** Yes, this is correct, the conversion is related to the vertical structure. We added two more sentences to further clarify what this quantity means but because it is a novel concept and several aspects would require extended explanations, we refer the reader to Schemm and Rivière (2019) for a more complete discussion.

**Reviewer:** Figure 7. The authors should clarify if the regions are kept similar in terms of longitudinal range and shifted latitudinally only, and if the area of the moving box is kept equal between the months or not. Also I would increase the black dots along the tracks and thicken the black box a bit to make them clearer to the eye.

**Authors:** The target region is kept similar in terms of longitudinal and latitudinal range and is shifted only in the meridional direction. We added this information to the main body of the manuscript and added the precise coordinates of the target region to the figure caption. We also increased the size of the black dots and the target box.

**Reviewer:** Figure 8 - looking at the figure I think the important feature for the midwinter minimum is the the *difference* between the before and after max-deepening which varies significantly between the months. The changes between Nov-Jan-Mar in the values before, or the values after the max deepening are not that large - specifically line 350- "The midwinter suppression affects only time steps after maximum deepening (gray boxes in Fig. 8)." is confusing because the Nov-Jan increase in the before box is larger than the Nov-Jan decrease in the after box, but the drop

in baroclinic conversion is much larger and I think this is what makes the difference because the EKE values in the northern box result from an integration over the whole eddy life cycle.

**Authors:** Yes, we agree and corrected this sentence accordingly. The drop is significant here and is now highlighted in the manuscript.

**Reviewer:** Sentence on lines 175-176 is very confusing.

**Authors:** We reformulated the sentence.